# Rural active aging as a development strategy: Nexus of physical exercise participation and multidimensional poverty in China

Qifei Xia[1], Shu Xuan[2], Guoyou Qin[3]*

1 School of Physical Education, Ankang College, Ankang, China, 2 School of Physical Education, Hunan University, Changsha, China, 3 College of Physical Education, Hanjiang Normal University, Shiyan, China

* qinguoyou22@163.com

## Abstract

### Background

This study aims to assess the intrinsic links between physical exercise and multidimensional poverty among rural older people, focusing on the mediating mechanisms of social and cultural capital, to inform responses to active ageing.

### Methods

This study utilized the adult and household databases of the China Household Tracking Survey (CFPS) in 2018 and 2022. Stata17.0 was used to screen and process the data, and the A-F double critical value method was adopted to measure and structurally decompose the multi-dimensional poverty among the elderly of Chinese household farmers. The association between physical exercise participation and multi-dimensional poverty among rural elderly people was estimated using the ordinary Least squares (OLS) method. The mediating effects of social capital and cultural capital were explored by using the stepwise regression method. Finally, the heterogeneous manifestations of the impact of physical exercise participation on multi-dimensional poverty among rural elderly people were investigated by using the grouped regression method.

### Results

This study found that physical exercise participation has a significant role in reducing multidimensional poverty in old age among family farmers ($\beta = -0.0149$, $p < 0.001$), and this finding remains valid after a series of robustness tests such as full model estimation ($\beta = -0.009$, $p < 0.001$), propensity score matching test (ATT value of $-0.017$, $-0.015$, and $-0.017$, respectively) and other series of robustness tests, the conclusion still holds. The results of the mediating effect showed that social capital and cultural capital mediated the effect of physical exercise on multidimensional

**Data availability statement:** "Publicly available datasets were analyzed in this study. This data can be found here: https://www.isss.pku.edu.cn/cfps/ (accessed on January 12, 2025). All relevant data can also be found in the supporting file: https://doi.org/10.6084/m9.figshare.29146913."

**Funding:** The author(s) received no specific funding for this work.

**Competing interests:** The authors have declared that no competing interests exist.

poverty. Heterogeneity analyses showed that the effects of physical exercise participation on multidimensional poverty were more pronounced among females, unmarried, in the eastern region, and among the elderly with a high life expectancy.

## Conclusion

Participation in physical exercise can effectively alleviate multidimensional poverty, which is reflected in the enhancement of health level, economic status, living standard, and subjective perception sub-dimensions. Engaging in physical exercise can alleviate poverty through two channels: accumulating social capital and enhancing cultural capital. The construction and maintenance of rural sports facilities should be strengthened, a 'healthy labour model' combining physical exercise and agricultural production should be constructed, special sports poverty-alleviation initiatives should be implemented for "vulnerable" elderly people, and a 'Healthy Villages' project should be launched to promote the participation of all people in poverty alleviation. The 'Healthy Villages' programme has been launched to promote the participation of the entire population in physical exercise.

## 1. Introduction

The eradication of poverty remains a central global issue and a critical prerequisite for achieving sustainable development. As the world's most populous developing country, China achieved a comprehensive victory in its poverty alleviation efforts by 2020. Under the current standards, all 98.99 million rural poor have been lifted out of poverty, 832 impoverished counties have been removed from the list, and 128,000 poor villages have been eliminated, effectively addressing regional poverty [1]. With the successful completion of poverty alleviation and the realization of a moderately prosperous society, absolute poverty under current standards has essentially been eliminated. However, poverty under alternative criteria persists and will remain a long-term challenge [2]. At this stage, China's poverty governance is shifting from addressing absolute poverty to tackling multidimensional poverty [3]. Rural elderly populations, particularly those with high poverty rates [4], face increasingly complex multidimensional poverty, which extends beyond income to encompass issues such as education, health, and life satisfaction [5]. These challenges are further compounded by various uncertainties, which directly affect the progress and effectiveness of rural poverty governance. As a result, consolidating the gains of poverty alleviation, reducing the risk of large-scale re-poverty among rural elderly populations, and accelerating comprehensive rural revitalization have become key focal points for both society and academia.

Multidimensional poverty refers to a poverty state determined by considering multiple welfare dimensions, including income, education, health, and housing, at the individual or household level [6]. In 2021, the proportion of elderly individuals aged 65 and above in China surpassed 14%, marking the country's official transition to a deeply aging society [7]. As a relatively vulnerable group, rural elderly face the dual challenges

of rapid aging and social imbalance, which has led to persistent multidimensional poverty. Subjectively, many rural elderly experience declining labor capacity, reduced adaptability, and diminishing social status, which makes them vulnerable to physical frailty due to illness or unexpected events, exacerbating their poverty [2]. Objectively, the urban-rural divide results in an underdeveloped rural social welfare system, with delayed services in healthcare, elderly care, and nursing, placing additional pressure on rural elderly who already face the dual burden of aging and poverty [8]. Thus, multidimensional poverty among rural elderly has become an urgent issue that must be addressed for the success of rural revitalization strategies [9]. With the implementation of the National Fitness Initiative and the "Healthy China" strategy, physical exercise has been increasingly promoted as a lifelong practice essential for enhancing physical fitness, improving quality of life, and fostering holistic development. The Healthy China 2030 Planning Outline, issued by the State Council, set a national target of 530 million regular participants in physical activity by 2023 [10]. However, in the context of persistent urban–rural economic disparities in China, the development of rural sports infrastructure lags significantly behind that of urban areas. Exploring the relationship between physical activity and multidimensional poverty among the rural elderly is thus critical to advancing the national strategy for active ageing. On one hand, exercise promotes public well-being, enhances quality of life [11], supports preventive health management, and reduces medical expenditures [12], thereby increasing disposable income and enabling dignified ageing [13]. On the other hand, existing studies suggest that physical activity improves rural residents' physical health, social capital, and psychological resilience [14]. It also helps maintain vitality, agency, social engagement, and independent living [15], ultimately contributing to an improved standard of living.

More importantly, for rural older people, physical exercise itself is a profoundly social practice and cultural activity. According to Bourdieu's theoretical framework [16], an individual's position in social space and his or her likelihood of escaping poverty are closely related to the social capital (the sum of actual or potential resources embedded in social networks) and cultural capital (internalised knowledge, skills, cultural literacy, and cognitive patterns) he or she possesses. The nature of multidimensional poverty in rural old age is often characterised by weak social network support and lack of access to resources (lack of social capital) as well as backward knowledge and skills, and insufficient awareness of health and rights (lack of cultural capital) [5]. Physical exercise provides a unique field for reshaping these two core capitals: as a vehicle for social interaction: participation in physical exercise effectively expands the social network of rural older adults, rebuilds neighbourhood mutual support, and promotes information exchange and emotional support [17]. The accumulation of such social capital has been shown to significantly enhance individuals' ability to obtain support [18], increase income stability [19], improve health [20], and increase life satisfaction [21], thereby systematically alleviating multidimensional poverty. In addition, differences in resource endowment and level of development across regions may also lead to heterogeneity in physical exercise participation and its poverty reduction effects.

To address these gaps, this study uses data from the China Family Panel Studies (CFPS) and applies the Alkire-Foster (A-F) dual cutoff method to measure the multidimensional poverty index of elderly members in rural households. The analysis is conducted at the household level to investigate the impact and mechanisms of physical exercise on multidimensional poverty. Our findings aim to offer both theoretical insights and empirical evidence to support long-term poverty alleviation and rural revitalization.

## 2. Literature review and research hypothesis

### 2.1. Multidimensional poverty measurement

As China transitions from high-speed growth to high-quality development, poverty governance has evolved from absolute poverty to multidimensional poverty. Absolute poverty refers to a state where individuals lack the basic necessities of life, such as essential goods and income. In contrast, relative poverty occurs when individuals or households, despite meeting basic material needs, live below the standard of their peers, placing them in a disadvantaged position [22]. Multidimensional poverty, on the other hand, is measured through a set of indicators that collectively reflect an individual's or household's overall well-being. A higher number of deprivations typically defines poverty. Since 2010, the United

 

Nations Development Programme's Human Development Report Office (HDRO) and the Oxford Poverty and Human Development Initiative (OPHI) have introduced the "Global Multidimensional Poverty Index (MPI)," which systematically captures the multidimensional nature of poverty, including health, education, and living standards. Amartya Sen's capability approach [23] further emphasizes the loss of function, such as the inability to avoid hunger, disease, or the failure to meet basic educational and social participation needs, as the root causes of poverty.In terms of measurement, two main approaches dominate the academic discourse. The first, marginal distribution methods like the Dashboard Approach, treat each dimension of poverty independently, neglecting interrelations. The second, proposed by Alkire and Foster [6], introduces a dual-threshold identification and measurement method. This method builds on the traditional FGT approach, overcoming the limitations of unidimensional poverty metrics through decomposition and identification indices.Recent studies frequently adopt the A-F method to assess multidimensional poverty among the elderly. For instance, Wang et al. [2] used CFPS data to measure multidimensional poverty in rural China, categorizing the MPI model into five dimensions: health, healthcare, education, income, and basic living standards. Zeng et al. [4] developed a relative poverty index based on four dimensions—health, social engagement, psychological well-being, and material needs—tailored to health-focused poverty alleviation policies (MRPI). This study follows the A-F method [2], incorporating health, economic status, living standards, and subjective well-being to establish a multidimensional poverty evaluation system. Using CFPS data, we decompose and aggregate poverty across these dimensions to calculate the multidimensional poverty index for elderly rural households.

## 2.2. Direct effects of physical exercise participation on multidimensional poverty among elderly rural households

In the post-poverty alleviation era, China's rural poverty reduction efforts have shifted from achieving the "two guarantees and three protections" to addressing and mitigating imbalances and insufficiencies in multidimensional poverty [24]. Drawing on Amartya Sen's capability approach, this study conceptualizes poverty not merely as a lack of income, but as a deprivation of the freedoms to achieve valuable functionings—such as good health, stable income, a quality life, and a sense of self-worth [6]. This framework centres on three interrelated components: capabilities, functionings, and conversion factors [25–28]. Capabilities refer to the real opportunities individuals have to achieve outcomes they value; functionings are the realized states or activities, such as being employed, engaging in community life, or maintaining good health; conversion factors influence how resources and opportunities are transformed into actual achievements. Physical exercise takes various forms and, based on its mode of organization, can be categorized into individual and group-based exercise. As a low-cost investment in individual health with wide-ranging benefits, individual participation in physical exercise offers inherent conditions and real possibilities for the alleviation of multidimensional poverty through the direct expansion and enhancement of personal viability. Specifically, it can contribute through several pathways.

1. **Health as a critical factor:** health plays a foundational role in labor capacity. According to human capital theory, health constitutes a critical form of human capital, and maintaining good physical condition is a prerequisite for engaging in productive work and generating income [29]. For rural elderly, factors such as increased population mobility, smaller family sizes, inadequate caregiving from children, and low income exacerbate poverty, creating a vicious cycle of "poverty leading to illness and illness leading to poverty" [30]. Individual participation in physical exercise maintains the ability of older people to take care of themselves [31], facilitates social interaction [32], and reduces the risk of disability [33], thus effectively safeguarding their level of health, which is the basis for engaging in productive activities, generating income, or reducing healthcare expenditures.

2. **Income effects**: The "trickle-down" effect suggests that economic growth has a poverty-reducing impact [34]. Physical exercise has a significant income premium, and good physical exercise habits of individuals contribute to better health and indirectly to income growth through enhanced labour force capacity [13]. Studies have shown that good health reduces the cost of healthcare in old age and supports a more stable source of income for farmers [35].

3. **Improved quality of life**: Physical exercise is a cost-effective means of enhancing cognitive function, improving quality of life, and reducing health disparities [36]. Participation in diverse physical activities helps mitigate negative attitudes toward aging [37], lowers frailty risks [38], and strengthens social identity [39], thereby enhancing living standards.

4. **Subjective well-being**: Compared with urban areas, rural older people at the medium level of development are gradually getting out of heavy agricultural production activities [40], and for some poor rural families, due to the frequent mobility of young labourers, the lack of children's financial support, and the impact of the negative events of widowhood, etc., participation in personal physical exercise activities (e.g., morning exercise, exercise with simple equipment) provides a positive way of relaxation and a channel of stress release, which can enhance their self-confidence in life and improve their subjective well-being, their self-confidence in life and enhance their subjective well-being [41].

Based on these considerations, we hypothesize:

- **H1**: Participation in physical exercise reduces multidimensional poverty among older rural individuals.

## 2.3. The mediating effects of social and cultural capital

Social and cultural capital may function as critical conversion factors in the poverty-alleviation effects of physical exercise. In the 1970s, Bourdieu's work introduced the concept of "social capital" in social sciences, defining it as the value of an individual's position within organizational structures [16]. Regular physical exercise fosters interaction and trust among rural residents, promoting strong social ties and improving quality of life [14]. For elderly rural populations, health security relies largely on traditional social networks such as family support and neighborly assistance [42]. Participation in physical activities enhances interpersonal skills, deepens mutual trust, and strengthens community engagement [43], facilitating access to social capital. Additionally, the accumulation of social capital plays a significant role in poverty reduction. Pham et al. [44] found that social networks enable farmers to access scarce resources, promote information exchange, and encourage deeper cooperation, thus reducing the risk of recidivism in poverty. It can be hypothesized that exercise strengthens internal ties and solidarity within rural communities, increasing social interaction among the elderly and mitigating issues of social exclusion and information poverty.

Cultural capital may also mediate the relationship between exercise participation and multidimensional poverty in elderly farming households. Bourdieu's theory of cultural capital [45] offers a theoretical framework, defining it as the precondition for engaging in aesthetic and recreational activities, closely tied to an individual's educational background. The accumulation of cultural capital typically manifests through cultural literacy, education, and refinement [46]. As both a social and cultural activity, exercise encourages participants to focus on knowledge acquisition and cultural development, leading to positive subjective experiences [47]. Research indicates that elderly individuals from various social strata accumulate cultural capital related to health and aging through participation in public physical activities [48]. Cultural poverty alleviation plays a crucial role in enriching the spiritual lives of impoverished populations and promoting sustainable poverty reduction. For rural residents in poverty, cultural and structural factors predominantly contribute to their economic challenges, with limited cultural capital, outdated cultural perspectives, and poor cultural exchange being key obstacles to wealth generation [49]. Enhancing family cultural capital supports deeper poverty alleviation and reduces the risk of reversion to poverty [50]. Based on this, the following hypotheses are proposed:

- **H2a**: Participation in physical exercise contributes to the accumulation of social capital, thereby reducing the likelihood of multidimensional poverty in elderly farming households.

- **H2b**: Participation in physical exercise enhances cultural capital, thereby reducing the likelihood of multidimensional poverty in elderly farming households.

Based on this literature review and hypothesis development, a theoretical framework model is proposed to explore the impact of physical exercise participation on multidimensional poverty among elderly farming households (see Fig 1).

## 3. Research design and data sources

### 3.1. Data sources

The data for this study primarily come from the China Family Panel Studies (CFPS) conducted by Peking University. This database provides comprehensive information on individuals, families, and communities, covering various areas such as socioeconomics, educational outcomes, and family relations. The CFPS has been updated through 2022, and this study uses data from the 2018 and 2022 waves to ensure timeliness. The sample includes rural elderly individuals aged 45 and above, with personal and family databases merged using family sample identification codes. Relevant variables were selected, and the data were cleaned, resulting in a final sample of 6,223 elderly rural households, with 3,876 from 2018 and 2,347 from 2022.

### 3.2. Variable selection

**3.2.1. Dependent variable.** The dependent variable is the multidimensional poverty (MPI) of elderly rural individuals, measured using the constructed Rural Elderly Multidimensional Poverty Index. The formula for calculating MPI is as follows: , $MPI \in [0, 1]$, where a higher MPI value closer to 1 indicates more severe multidimensional poverty among elderly rural households.

Currently, there is no unified standard for calculating the Multidimensional Poverty Index (MPI) both domestically and internationally. To ensure the operational feasibility of this study, we use the "Global Multidimensional Poverty Index (MPI)" as the reference framework. Based on the work of Song et al. [51], four dimensions—health status, economic conditions, living standards, and subjective well-being—are selected, incorporating 12 specific indicators to construct the multidimensional poverty index system. Health status reflects the physical and mental health of elderly rural individuals. Given the significant missing values in health data for 2022, subjective evaluations were used as a proxy. Economic condition serves as a direct measure of poverty, while living standards assess material deprivation in the elderly population. Subjective well-being considers individuals' overall evaluation of their quality of life. Equal weighting is applied across the four dimensions, with branch indicators within each dimension averaged to calculate the final MPI. The set multidimensional poverty index system is shown in Table 1.

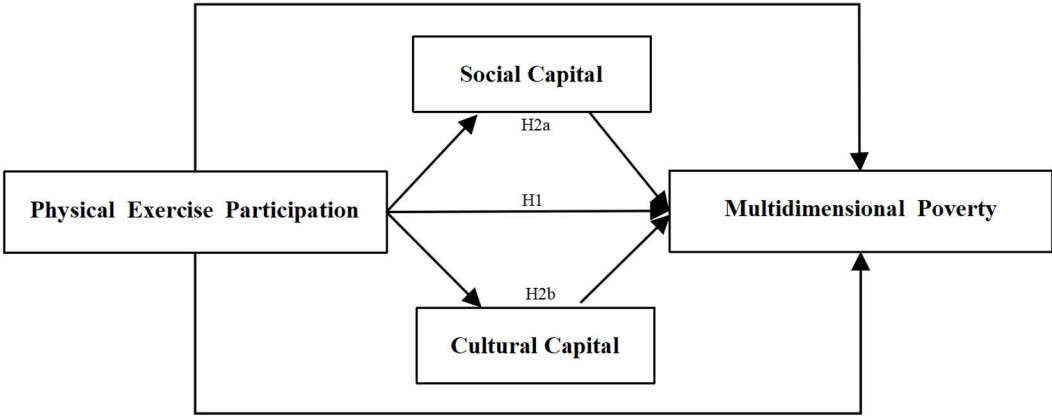

**Fig 1. Theoretical framework model.**

**Table 1. Multidimensional poverty dimensions, indicators and thresholds for older adults.**

| Dimension (Weight) | Indicator (Weight) | CFPS Survey Question and Response Options | Deprivation Threshold |
|---|---|---|---|
| **Health Status** (1/4) | Self-rated Health (1/16) | Q: "How would you rate your overall health status?" Response: Scale 1–5 (Very healthy to Unhealthy) | Deprived if score = 5 (coded as 1) |
| | Physical Functionality (1/16) | Q: "Which activities can you not perform independently?" Response: ADL Scale including: 1) Outdoor activities; 2) Eating; 3) Cooking; 4) Using public transport; 5) Shopping; 6) Cleaning; 7) Laundry | Deprived if unable to perform any one activity independently (coded as 1) |
| | Depression Status (1/16) | Q: "How often have you felt depressed or unmotivated in the past week/month?" Response: 1) Almost never; 2) Sometimes; 3) Often; 4) Most of the time | Deprived if response > 2 (coded as 1) |
| **Economic Status** (1/4) | Housing Ownership (1/8) | Q: "Who owns your current residence?" Response: 1) Full ownership by family; 2) Partial ownership; 3) Public housing; 4) Affordable housing; 5) Public rental; 6) Commercial rental; 7) Relatives' property | Deprived if response > 2 (coded as 1) |
| | Per Capita Consumption (1/8) | Calculated using World Bank PPP conversion factor (2017): Household expenditure/(household size) | Deprived if adjusted per capita consumption < $1.25/person/day (coded as 1) |
| **Living Standards** (1/4) | Access to Water (1/16) | Q: "What is your primary water source for cooking?" Response: 1) River/lake; 2) Well; 3) Tap water; 4) Bottled/purified; 5) Rain; 6) Cistern; 7) Pond/spring; 77) Other | Deprived if not using options 3 or 4 (coded as 1) |
| | Cooking Fuel (1/16) | Q: "What is your primary cooking fuel?" Response: 1) Firewood; 2) Coal; 3) LPG; 4) Natural gas; 5) Solar/biogas; 6) Electricity; 7) Other | Deprived if not using options 3, 4, 5, or 6 (coded as 1) |
| | Electricity Access (1/16) | Q: "What is your household's electricity situation?" Response: 1) No access; 2) Frequent outages; 3) Occasional outages; 4) Almost no outages | Deprived if response = 1 (coded as 1) |
| | Durable Goods (1/16) | Q: "Do you own any durable consumer goods?" (Items valued > ¥1,000 with >2-year lifespan) Response: Yes/No | Deprived if no durable goods owned (coded as 1) |
| **Subjective Well-being** (1/4) | Life Satisfaction (1/8) | Q: "How satisfied are you with your life?" Response: Scale 1–5 (Very dissatisfied to Very satisfied) | Deprived if response < 3 (coded as 1) |
| | Future Confidence (1/8) | Q: "How confident are you about your future?" Response: 1) Very unconfident; 2) Unconfident; 3) Neutral; 4) Confident; 5) Very confident | Deprived if response < 3 (coded as 1) |

**(1) Method for multidimensional poverty measurement.**

To ensure the fairness and scientific rigor of the multidimensional poverty assessment, this study employs the dual-cutoff method (A-F method) first proposed by Alkire and Foster [6], to evaluate the multidimensional poverty of elderly rural households. This approach aggregates dimensions to calculate a composite poverty index for each dimension, enabling precise measurement of the incidence of multidimensional poverty (H), the average deprivation share (A), and the Multidimensional Poverty Index (MPI) for households or individuals. The process is outlined as follows:

Assume that a total of $N$ households are surveyed, where $i$ ($i \in N$) represents the $i$ household in the sample. Each household is assessed across $d$ poverty dimensions, with $j$ ($j \in d$) denoting the $j$ dimension. Let $g_{ij}$ represent the observed value of the $i$ household for the $j$ dimension, and $z$ be the poverty threshold for dimension $j$. The poverty status $P_{ij}$ of the $i$ household in the $j$ dimension is defined as:

$$P_{ij} = \begin{cases} 1, & \text{if } g_{ij} \le z_j \\ 0 & \text{Other Conditions} \end{cases} \tag{1}$$

$$r_{ij} = p_{ij} * w_j \tag{2}$$

Here, $wj$ represents the weight for dimension $j$. By assigning weights to each poverty dimension, the weighted deprivation value for each dimension can be derived. Multidimensional poverty is then identified by the number of deprived dimensions $k$ ($k \in d$):

$$c_{ij}(k) \begin{cases} \sum_{j=1}^{d} r_{ij}, & if \sum_{j=1}^{d} r_{ij} \geq k \\ 0, & if \sum_{j=1}^{d} r_{ij} < k \end{cases} \tag{3}$$

From this, the deprivation share for each value of $k$ can be derived, followed by the identification of the number of individuals experiencing multidimensional poverty for each $k$-value:

$$q_{ij}(k) \begin{cases} 1, & if c_{ij}(k) > 0 \\ 0, & if c_{ij}(k) = 0 \end{cases} \tag{4}$$

Subsequently, by summing the deprivation shares, the Multidimensional Poverty Index (MPI) can be calculated, yielding the multidimensional poverty incidence H(k):

$$H(k) = \frac{\sum_{i=1}^{n} q_{ij}(k)}{N} \tag{5}$$

The deprivation share A($k$) is given by:

$$A(k) = \frac{\sum_{i=1}^{n} c_{ij}(k)}{\sum_{i=1}^{N} q_{ij}(k) * d} \tag{6}$$

The Multidimensional Poverty Index M($k$) is given by:

$$M(k) = H(k) * A(k) \frac{\sum_{i=1}^{n} c_{ij}(k)}{N * d} \tag{7}$$

## (2) Calculation results

Based on the multidimensional poverty thresholds defined by the United Nations, we calculated the overall poverty thresholds ranging from 0.1 to 0.4, with a k-value ≥ 0.3 indicating multidimensional poverty and a k-value ≥ 0.4 denoting severe multidimensional poverty. Using the multidimensional poverty index for elderly rural households in China for 2018 and 2022 (see Table 2), the results (Fig 2) reveal that as the k-value increases, the incidence and index of multidimensional poverty decline, ultimately approaching zero. These findings suggest a significant reduction in multidimensional poverty among elderly rural households in China, reflecting notable progress in poverty alleviation efforts.

**3.2.2. Explanatory variables.** The core explanatory variable is participation in physical exercise among elderly individuals in rural Chinese households. The CFPS dataset captures a wide range of physical activities, including periodic exercises such as walking, running, jogging, and hiking; ball games such as football, basketball, and table tennis; and water-based activities like swimming, sailing, and diving. According to prior research [52], total physical activity should be assessed across dimensions of frequency, duration, and intensity. The American College of Sports Medicine (ACSM) recommends engaging in at least three sessions of moderate-intensity physical activity per week, each lasting 30 minutes or more [53]. In this study, participation in physical exercise is proxied by exercise frequency. Due to substantial missing and abnormal values in the exercise duration variable across CFPS waves, we use self-reported frequency as the indicator. Based on the survey question "How many times did you exercise in the past week?" and following Liu [14], we construct a binary variable: individuals reporting exercise ≥3 times per week are classified as participating in physical exercise (coded as 1); otherwise, they are coded as 0.

**Table 2. Results of multidimensional poverty measurement.**

| Year | K-value | Multidimensional Poverty Headcount Ratio (H) | Average Deprivation Share (A) | Multidimensional Poverty Index (MPI) |
|------|---------|----------------------------------------------|-------------------------------|--------------------------------------|
| 2018 | 0.1 | 0.525 (52.5%) | 0.191 (19.1%) | 0.100 (10.0%) |
|      | 0.2 | 0.146 (14.6%) | 0.300 (30.0%) | 0.043 (4.3%) |
|      | 0.3 | 0.070 (7.0%)  | 0.355 (35.5%) | 0.024 (2.4%) |
|      | 0.4 | 0.012 (1.2%)  | 0.460 (46.0%) | 0.005 (0.5%) |
| 2022 | 0.1 | 0.499 (49.9%) | 0.197 (19.7%) | 0.098 (9.8%) |
|      | 0.2 | 0.149 (14.9%) | 0.311 (31.1%) | 0.046 (4.6%) |
|      | 0.3 | 0.078 (7.8%)  | 0.366 (36.6%) | 0.028 (2.8%) |
|      | 0.4 | 0.016 (1.6%)  | 0.478 (47.8%) | 0.007 (0.7%) |

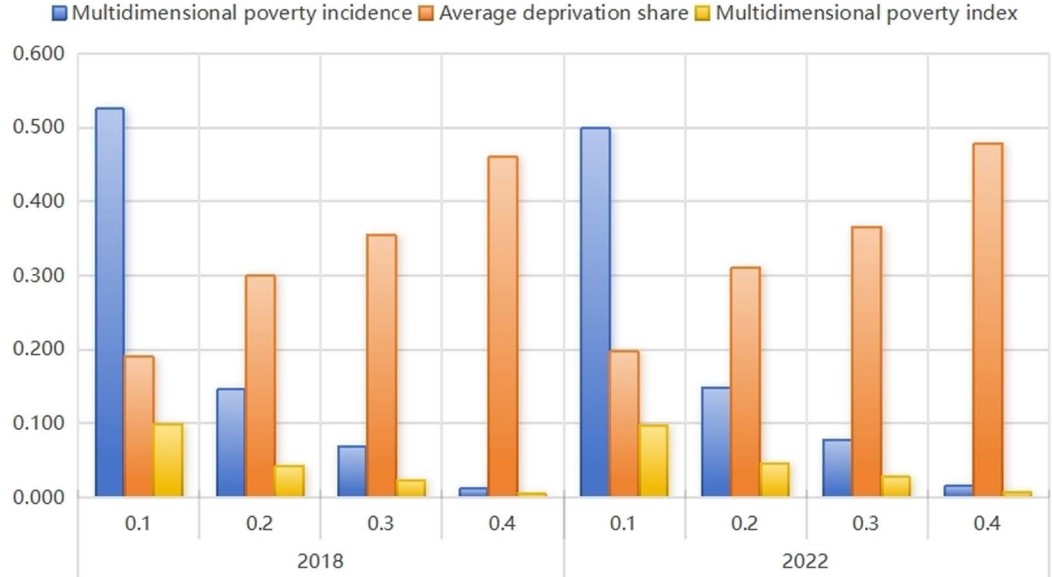

**Fig 2. Multidimensional poverty index for different values of *k*.**

**3.2.3. Mediating variables.** The first mediating variable examined in this study is social capital. Social capital can be conceptualized at three levels: micro, meso, and macro. At the micro level, it reflects interpersonal social networks and is referred to as individual social capital; at the meso and macro levels, it concerns individuals' engagement with collectives or institutions, thus representing collective social capital. Accordingly, this study measures social capital across both individual and collective dimensions.Following established approaches based on CFPS data [54], individual social capital is assessed through interpersonal relationship quality. Respondents were asked, "How good are your relationships with others?" with responses ranging from 0 to 10. To facilitate interpretation, responses below 5 were coded as 0 (indicating poor interpersonal relations), and responses of 5 or above were coded as 1 (indicating good interpersonal relations), thereby generating a binary indicator of individual social capital.For collective social capital, this study incorporates three components—reciprocity, trust, and organizational participation—based on prior literature [2,55,56]. These are operationalized using three CFPS survey items:(1) Reciprocity is measured by the question, "Do you think most people are helpful or selfish?" Responses indicating "most people are helpful" were coded as 1 (reciprocal), while those indicating "most people are selfish" were coded as 0 (non-reciprocal).(2) Trust is assessed via the question, "Do you generally trust

others or remain cautious?" Responses of "most people can be trusted" were coded as 1 (trusting), whereas "you can never be too careful" responses were coded as 0 (distrustful). (3) Organizational participation is captured through the question, "Which organizations are you a member of?" Respondents who reported participating in at least one political or social organization were coded as 1 (participant); others were coded as 0 (non-participant). Principal component analysis (PCA) was employed to compute a composite score for collective social capital across the three dimensions. Both individual and collective social capital measures were subsequently standardized and converted into binary variables for analytical consistency..

The second category of mediating variables is cultural capital. Bourdieu classifies cultural capital into embodied, objectified, and institutional forms [45]. Due to limited data on objectified and institutional cultural capital in CFPS, we focus on individual and institutional cultural capital. Individual cultural capital is measured by the question: "In the past 12 months, have you read books or newspapers?" Responses of "Yes" are coded as 1, and "No" as 0. Educational level, representing institutional cultural capital, is used as a proxy. The CFPS question is: "What is your highest level of education?" Answers are coded as follows: illiterate/semi-illiterate/primary school = 1, middle school/high school/vocational school = 2, college/university/graduate degree = 3. The arithmetic average of individual and institutional cultural capital is then computed.

**3.2.4. Control variables.** To ensure the accuracy of the analysis on the impact of exercise participation on multidimensional poverty among elderly rural households, and to mitigate sample selection bias and omitted variable issues, individual and regional characteristics of household heads are included as control variables. Individual characteristics include gender, age, and marital status, with gender and marital status coded as binary variables. Regional characteristics are categorized based on geographic location and economic development, distinguishing between the central-western and eastern regions. Descriptive statistics for these variables are provided in Table 3.

## 3.3. Analytical Strategy

Ordinary Least Squares (OLS) is a linear least squares method used to estimate unknown parameters in a linear regression model. The results estimated by OLS regression model in the case of large samples are more intuitive, economically meaningful, and convenient for presenting the marginal effects of the explanatory variables on the explained variables. Since the dependent variable multidimensional poverty in this study is a continuous variable and the explanatory variable is a categorical variable, the following multiple linear regression model was set up in order to validate the research hypotheses presented in the previous section and to accurately identify the effects of physical exercise participation on multidimensional poverty in old age among family farmers:

$$MPI_i = \alpha_0 + \alpha_1 PA_i + \alpha_2 X_i + \delta_i \qquad (1)$$

**Table 3. Results of descriptive statistics for each variable.**

| Variable Type | Variable Name | Description | Mean | Std. Dev. |
|---|---|---|---|---|
| **Dependent Variable** | Multidimensional Poverty Status | Calculated using A-F indicators | 0.115 | 0.102 |
| **Independent Variable** | Participation in Physical Exercise | 0 = Non-participant, 1 = Participant | 0.672 | 0.469 |
| **Mediating Variables** | Social Capital | 0 = Low social capital, 1 = High social capital | 0.982 | 0.129 |
| | Cultural Capital | Arithmetic mean of personal and institutional cultural capital | 0.677 | 0.467 |
| **Control Variables** | Household Head Gender | 0 = Female, 1 = Male | 0.546 | 0.497 |
| | Household Head Age | Years (calculated from reference year) | 57.849 | 8.980 |
| | Household Head Marital Status | 0 = Single, 1 = Married | 0.858 | 0.348 |
| | Region | 0 = Central/Western, 1 = Eastern | 0.352 | 0.477 |
| **Instrumental Variable** | Provincial Investment in Public Fitness Centers | Logarithmic transformation | 8.675 | 1.148 |

In Equation (1), $MPI_i$ represents the dependent variable, indicating the multidimensional poverty status of elderly rural households. $PA_i$ is the independent variable, denoting exercise participation, and $X_i$ represents a series of control variables, such as gender and age. $\delta_i$ is the random error term.

To examine the mechanism of physical exercise participation on multidimensional poverty in old age among family farmers, we constructed a mediation effect model diagram (see Fig 3). Where physical activity is the core explanatory variable and MPI is the explanatory variable. Bourdieu's capital theory of social capital (relationship network) and cultural capital (knowledge and skills) are two independent dimensions with different accumulation paths and do not necessarily have a transformational relationship, so social capital and cultural capital are tested as two independent mediating variables. In the model diagram, firstly, it is necessary to judge whether physical exercise has a direct effect on MPI, and if there is a significant effect, then the mediating effect of social capital and cultural capital is further verified by stepwise regression.

To investigate the mechanism through which exercise participation impacts multidimensional poverty among elderly rural households, the following mediating effect model is constructed based on the stepwise regression method:

1. The first step is to examine the main effect of exercise participation on multidimensional poverty among rural households, as specified in Model (1).

2. The second step is to examine the indirect effect of exercise participation on multidimensional poverty. The model is specified as follows:

$$Mediations_i = \gamma_0 + \gamma_1 MPI_i + \gamma_2 X_i + \delta \tag{2}$$

In Equation (2), $Mediations_i$ represents the mediating variables, specifically social capital and cultural capital, while $MPI_i$ denotes the multidimensional poverty status of elderly rural households.

3. The third step is to test the validity of the mediating mechanism, as specified in the following model:

$$MPI_i = \eta_0 + \eta_1 MPI_i + \eta_2 Mediations_i + \eta_3 X_i + \delta \tag{3}$$

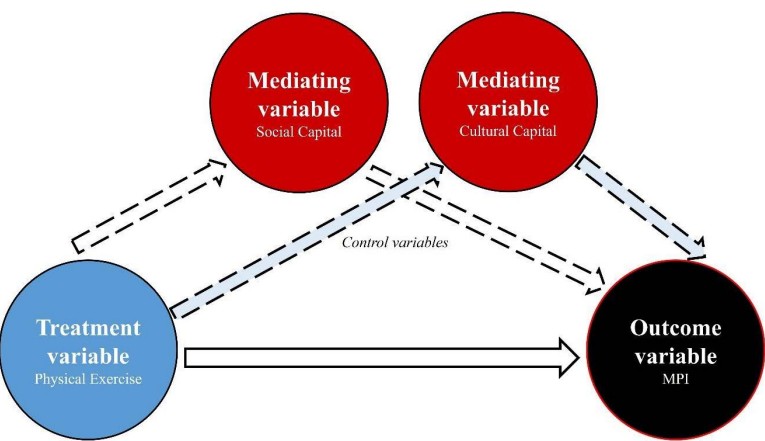

**Fig 3. Modelled diagram of the mediation effect.**

In Equation (3), $MPI_i$ represents the multidimensional poverty status of elderly rural households. A significant coefficient $\alpha_1$, along with a significant product $\gamma_1 \times \eta_2$, indicates the presence of a mediating effect. Conversely, if α1 is not significant, or if $\gamma_1 \times \eta_2$ is not significant, no mediating effect is considered to exist.

## 4. Empirical results and analysis

### 4.1. OLS model regression results

Multiple linear regression analysis (OLS model) was performed using Stata 17.0 statistical software. Model (1) is the result of regression without including any characteristics, and the regression result shows that the coefficient of physical activity participation is −0.008, which is significant at 1% level condition, i.e., for every unit increase in physical activity participation, the multidimensional poverty of family farmers in old age decreases by 0.008 percentage points, which indicates that the more the tendency to participate in physical activity in old age of the family farmers, the more pronounced is the effect of poverty reduction. Research evidence suggests that long-term participation in physical activity develops an individual's ability to better cope with daily life, sense of group belonging and social self-confidence, as well as the ability to positively shape physical and mental change to better address poverty [16]. Thus, our study also yielded consistent results demonstrating the poverty-reducing effects of physical activity. Model (2) incorporates individual characteristics and the effect of physical activity participation on multidimensional poverty in old age among family farmers remains significantly different ($\beta = -0.0146$, $p < 0.001$), i.e., it shows that by controlling for other variables, participation in physical activity significantly reduces the multidimensional poverty index by 0.0146 units. Individual-level gender, age, and marriage all significantly affect multidimensional poverty. Model (3) incorporates regional characteristics, and the results show that region does not have a significant effect on multidimensional poverty, but the direction of the regression is shown to be negative, i.e., it suggests that family farmers in the less-developed regions of the west may face more severe multidimensional poverty in old age. In order to verify the reliability of the results, the OLS model regression was conducted for different years respectively, and the relevant features were controlled in the model validation, and the results of model (4) and model (5) showed that the poverty reduction effect of physical activity increased year by year ($\beta = -0.0171$, $p < 0.001$; $\beta = -0.0508$, $p < 0.05$), which further verified the robustness of the results. See Table 4.

### 3.2. Mechanism testing

#### 3.2.1. Mediation effect testing.
According to the mediating effect model diagram, in order to verify whether social capital and cultural capital have a mediating effect in physical exercise participation affecting multidimensional poverty in old age among family farmers, we used a hierarchical stepwise regression to conduct the test. In Model (1), physical exercise shows a significant negative association with multidimensional poverty ($\beta = -0.0149$, $p < 0.001$). In Models (2) and (4), the coefficients for social capital and cultural capital are both significantly positive, indicating that exercise participation promotes the accumulation of these forms of capital among elderly rural residents. Model (3) reveals a significantly negative coefficient, suggesting that physical activity alleviates poverty through enhanced social capital. Prior studies indicate that even informal or leisure-based physical activities can help individuals expand their social capital [57], enabling rural households to access poverty-reduction resources through broader social networks, thereby reducing the risk of poverty recurrence [2]. Similarly, Model (5) shows a significant negative coefficient, indicating that exercise mitigates multidimensional poverty through the enhancement of cultural capital. Evidence suggests that individuals who regularly engage in exercise tend to acquire more health-related knowledge and develop stronger health values and behavioral norms—key components of cultural capital [58]. When cognitive patterns, habitual behaviors, and resource access capabilities reach a certain threshold, they may help break the intergenerational transmission of poverty [50] (see Table 5).

**Table 4. Benchmark model regression results.**

| Variables | Model (1) | Model (2) | Model (3) | Model (4) | Model (5) |
|---|---|---|---|---|---|
| Sports Participation | −0.00854*** | −0.0146*** | −0.0149*** | −0.0171*** | −0.0508** |
| | (0.00277) | (0.00273) | (0.00273) | (0.00317) | (0.0208) |
| Gender (Male = 1) | | −0.0119*** | −0.0121*** | −0.0113*** | −0.0144*** |
| | | (0.00254) | (0.00254) | (0.00312) | (0.00433) |
| Age | | 0.00211*** | 0.00212*** | 0.00213*** | 0.00217*** |
| | | (0.000147) | (0.000147) | (0.000188) | (0.000237) |
| Marital Status (Married = 1) | | −0.0417*** | −0.0415*** | −0.0376*** | −0.0496*** |
| | | (0.00373) | (0.00373) | (0.00456) | (0.00652) |
| Region (Eastern = 1) | | | −0.00443* | −0.00669** | 0.000770 |
| | | | (0.00263) | (0.00319) | (0.00464) |
| Constant | 0.121*** | 0.0459*** | 0.0470*** | 0.0433*** | 0.0889*** |
| | (0.00228) | (0.00971) | (0.00973) | (0.0124) | (0.0251) |
| Observations | 6223 | 6223 | 6223 | 3876 | 2347 |
| R-squared | 0.001 | 0.071 | 0.071 | 0.074 | 0.071 |

Note. Standard errors are given in parentheses.

* $p < 0.05$.

** $p < 0.01$.

*** $p < 0.001$.

**Table 5. Results of the mediation effect test.**

| Variables | (1) | (2) | (3) | (4) | (5) |
|---|---|---|---|---|---|
| | MPI | SC | MPI | CC | MPI |
| Sports Participation | −0.0149*** | 0.0200*** | −0.0132*** | 0.290*** | −0.00988*** |
| | (0.00273) | (0.00356) | (0.00272) | (0.0255) | (0.00272) |
| Social Capital | | | −0.0827*** | | |
| | | | (0.00967) | | |
| Cultural Capital | | | | | −0.0171*** |
| | | | | | (0.00134) |
| Constant | 0.0470*** | 1.065*** | 0.135*** | 2.511*** | 0.0901*** |
| | (0.00973) | (0.0127) | (0.0141) | (0.0909) | (0.0102) |
| Controls | Yes | Yes | Yes | Yes | Yes |
| Observations | 6223 | 6223 | 6223 | 6223 | 6223 |
| R-squared | 0.071 | 0.014 | 0.082 | 0.110 | 0.095 |

## 4.4. Robustness checks

To ensure the validity and reliability of the results regarding the impact of physical exercise participation on multidimensional poverty in elderly rural households, we conducted three robustness checks on the baseline model. **1. Full model estimation**: Following the approach of Liu et al. [14], we included two mediators—social capital and cultural capital—and two moderators—internet service use and public service levels—into the baseline model as control variables. The results from Model (1) show that the regression coefficient for physical exercise participation is −0.0134, significant at the

1% level. **2. Control for provincial and year fixed effects**: To account for potential biases from provincial or temporal factors, we controlled for fixed effects in both provinces and years. The results in Model (2) show a negative coefficient for physical exercise participation ($\beta = -0.0178$, $p < 0.01$), confirming the robustness of the findings. **3. Propensity score matching (PSM)**: To account for the potential non-random nature of physical exercise participation, and to mitigate selection bias in estimating its effect on multidimensional poverty among the rural elderly, this study employed propensity score matching (PSM). The advantage of the PSM method is that the control of confounders is shifted to the control of propensity values in order to 'downscale' and control the confounding bias. By matching individuals with similar propensity scores across treatment (exercise participation) and control groups, PSM enables estimation of the average treatment effect on the treated (ATT) while reducing selection bias. Three matching algorithms were used to assess robustness: nearest-neighbor matching, kernel matching, and radius matching (caliper = 0.01). As shown in Table 6, the estimated ATT values from the three methods were −0.017, −0.015, and −0.017, with corresponding t-values of −5.01, −5.34, and −5.64, all exceeding the critical threshold of 1.96 and statistically significant at the 1% level. These results confirm the robustness of the matching procedure and suggest that even after accounting for selection bias, physical activity participation has a significant negative effect on multidimensional poverty among elderly rural households (See Table 7) **4. Multidimensional poverty decomposition:** As shown in Table 8, Model indicates that physical exercise significantly improves health status, economic conditions, living standards, and subjective wellbeing. These results suggest that exercise contributes

**Table 6. ATT effects of propensity score matching.**

| Matching Method | ATT | Standard Deviation (SD) | t-value |
|---|---|---|---|
| Nearest-Neighbor Matching | −0.0177 | 0.0035 | −5.01 |
| Kernel Matching | −0.0158 | 0.0029 | −5.34 |
| Radius Matching | −0.0176 | 0.0031 | −5.64 |

**Table 7. Robustness test results.**

| | (1) | (2) | (3) |
|---|---|---|---|
| **Variables** | **Full Model** | **Province & Year Controls** | **PSM** |
| **Sports Participation** | −0.009*** | −0.0178*** | −0.0148*** |
| | (0.00273) | (0.00318) | (0.00273) |
| **Constant** | 0.179*** | 0.0470*** | 0.0452*** |
| | (0.0143) | (0.00972) | (0.0452) |
| **Controls** | Yes | Yes | Yes |
| **R-squared** | 0.106 | 0.071 | 0.070 |

**Table 8. Decomposition of multidimensional poverty.**

| | (1) | (2) | (3) | (4) |
|---|---|---|---|---|
| **Variables** | **Health status** | **Economic condition** | **Living standard** | **Subjective wellbeing** |
| Sports Participation | −0.0181*** | 0.0211*** | −0.0538*** | 0.0131*** |
| | (0.00661) | (0.00461) | (0.00594) | (0.00503) |
| Constant | 0.00689 | −0.0259 | 0.0912*** | 0.882*** |
| | (0.0236) | (0.0164) | (0.0212) | (0.0179) |
| Observations | 6223 | 6223 | 6223 | 6223 |
| R-squared | 0.064 | 0.025 | 0.038 | 0.008 |

to enhanced functional ability and reduced depressive symptoms, increased consumption, reduced daily living expenses, and improved self-perception. Collectively, these improvements further validate the poverty-reducing effects of physical activity.

### 4.3. Heterogeneity analysis

In order to further test the heterogeneity of physical exercise participation on different characteristics among the older age groups of family farmers, multivariate linear models were used to explore the differences across gender, marital status, region, and age, respectively.

1. **Gender:** Physical exercise participation significantly reduces multidimensional poverty in both male and female elderly rural households. However, the effect is more pronounced for women. A Seemingly Unrelated Regression (SUR) test for group differences revealed no significant difference between genders ($p = 0.294$). (See Table 9)

2. **Marital status**: The impact of physical exercise on multidimensional poverty is significant for both unmarried and married elderly individuals. However, the effect is more pronounced among unmarried elderly. The SUR test for group differences indicated a significant difference ($p = 0.016$), suggesting that physical exercise has the largest impact on unmarried elderly individuals. (See Table 9)

3. **Region**: Physical exercise participation significantly reduces multidimensional poverty in elderly rural households across all regions. However, the effect is stronger in the eastern regions ($p = 0.033$). Therefore, physical exercise has a more substantial poverty-reducing effect for elderly households in the eastern regions. (See Table 9).

4. **Age:** Following the World Health Organization's classification of age groups [59], individuals aged 45–59, 60–74, and 75 years and above were categorized as middle-aged, elderly, and oldest-old, respectively. As shown in Table 11, the effect of physical exercise on reducing multidimensional poverty was most pronounced among the oldest-old group. Seemingly unrelated regression (SUR) was used to test for intergroup differences. The results indicate no significant difference between the middle-aged and elderly groups ($p = 0.926$), whereas significant differences were observed between the elderly and the oldest-old ($p = 0.027$), and between the middle-aged and the oldest-old ($p = 0.021$). (See Table 10). To further verify the differential effect of the mediating effect in different age groups, age group regressions of the mediating effect were conducted separately using the OLS model and tested for between-group differences. The results were found. At the level of social capital, the transmission mechanism of physical activity affecting multidimensional poverty among rural elderly through social capital exists in different age groups of elderly, and the regression coefficient is larger in the high life elderly group, indicating that the high life elderly group can exert the poverty reduction effect of physical activity to a greater extent through social capital. After comparing the coefficients of difference between groups, it was found that there was no significant difference between the middle-aged group and the old-aged

**Table 9. Heterogeneity test results for gender, marriage and region.**

| Statistics | (1) Female | (2) Male | (3) Unmarried | (4) Married | (5) Central/Western | (6) Eastern |
|---|---|---|---|---|---|---|
| Sports Participation | −0.0176*** | −0.0120*** | −0.0331*** | −0.0119*** | −0.0121*** | −0.0237*** |
|  | (0.00416) | (0.00359) | (0.00841) | (0.00286) | (0.00340) | (0.00467) |
| Controls | Yes | Yes | Yes | Yes | Yes | Yes |
| Observations | 2823 | 3400 | 879 | 5344 | 4024 | 2199 |
| R-squared | 0.081 | 0.056 | 0.042 | 0.037 | 0.048 | 0.061 |
| *P* | 0.294 |  | 0.016** |  | 0.033** |  |

**Table 10. Heterogeneity of the Effect of Physical Exercise on Multidimensional Poverty Across Age Groups.**

| | (1) | (2) | (3) |
|---|---|---|---|
| Variables | Middle-aged adults | Older adults | Oldest-old adults |
| Physical Exercise Participation | −0.0112*** | −0.0118** | −0.0536*** |
| | (0.00313) | (0.00539) | (0.01792) |
| Control Variables | Yes | Yes | Yes |
| Observations | 3852 | 2059 | 312 |
| R-squared | 0.026 | 0.028 | 0.096 |
| P | 0.926、0.027**、0.021** | | |

**Table 11. Tests for age heterogeneity in the mediating effects of social and cultural capital.**

| | Social capital | | | Cultural capital | | |
|---|---|---|---|---|---|---|
| Variables | (1)<br>Middle-aged adults | (2)<br>Older adults | (3)<br>Oldest-old adults | (4)<br>Middle-aged adults | (5)<br>Older adults | (6)<br>Oldest-old adults |
| Physical Exercise Participation | −0.00897*** | −0.0114** | −0.0529*** | −0.00686** | −0.00665 | −0.0498*** |
| | (0.00314) | (0.00537) | (0.0175) | (0.00312) | (0.00535) | (0.0181) |
| Social capital | −0.0907*** | −0.0734*** | −0.109*** | | | |
| | (0.0156) | (0.0149) | (0.0266) | | | |
| Cultural capital | | | | −0.0164*** | −0.0195*** | −0.0108 |
| | | | | (0.00164) | (0.00242) | (0.00762) |
| Control Variables | Yes | Yes | Yes | Yes | Yes | Yes |
| Observations | 3852 | 2059 | 312 | 3852 | 2059 | 312 |
| R-squared | 0.036 | 0.042 | 0.154 | 0.053 | 0.060 | 0.113 |
| P | 0.695、0.018**、0.030** | | | 0.972、0.020**、0.023** | | |

group (P = 0.695), there was a significant difference between the middle-aged group and the old-aged group (P = 0.018), and there was a significant difference between the old-aged group and the high-lived old-aged group (P = 0.030). At the level of cultural capital, the transmission mechanism of physical activity affecting multidimensional poverty in rural old age through cultural capital was significantly different only in the middle-aged and high life old age groups, and the regression coefficient was greater in the high life old age group. After comparing the coefficient of variation between groups, it was found that there was no significant difference between the middle-aged group and the elderly group (P = 0.972), there was a significant difference between the middle-aged group and the elderly group (P = 0.020), and there was a significant difference between the elderly group and the high-lived elderly group (P = 0.023) (See Table 11).

## 5. Discussion

### 5.1. Impact of physical exercise participation on multidimensional poverty in elderly rural households

The findings of this study indicate that participation in physical exercise significantly reduces poverty among elderly rural households. Regular physical activity has been shown to effectively alleviate poverty. The 2023 Central Document No. 1, *Opinions on Advancing Rural Revitalization*, calls for strengthened monitoring and support measures to further improve the living standards of impoverished populations. Poverty alleviation extends beyond economic relief; it is a multifaceted social issue. In this context, physical exercise, as a low-cost and sustainable intervention, is emerging as a new direction for poverty alleviation, particularly in rural areas. On one hand, exercise improves physical health by enhancing immunity and cardiovascular function, reducing the economic burden associated with poor health [60]. On the other hand, moderate

exercise fosters social cohesion, improves mental health, and enhances the self-development and social participation of impoverished groups [61]. For elderly rural populations, regular exercise mitigates health risks associated with harsh living conditions, inadequate medical resources, and poor nutrition. It helps them navigate life's challenges and boosts their confidence in problem-solving. This psychological shift aids elderly individuals in adapting to social changes and enhancing their resilience to life's difficulties. Therefore, within the framework of rural revitalization, it is essential to develop a "sports + health" poverty reduction model. This model should focus on raising awareness among elderly rural households about the benefits of physical exercise and provide them with tailored exercise programs and health knowledge, contributing to more

## 5.2. Mediating roles of social and cultural capital

The mediation analysis reveals that physical exercise participation significantly impacts multidimensional poverty in elderly rural households through two primary pathways: the accumulation of social capital and the enhancement of cultural capital.

**Mediating role of social capital:** Physical exercise, particularly group activities such as walking or square dancing, provides elderly individuals in impoverished areas with opportunities for social interaction [62]. These social networks enable them to receive advice and support during health crises, alleviating feelings of isolation and securing necessary medical assistance, which helps reduce health-related poverty and improve subjective well-being. In rural areas, where elderly populations face challenges due to changing family structures, youth migration, and limited social services, social capital theory suggests that participation in physical activities strengthens social ties, facilitates access to support, and offers a viable pathway out of multidimensional poverty [63]. For instance, community-based fitness programs in rural areas not only enhance physical health but also build social capital, boosting mental well-being and productivity, which in turn improves disposable income.

**Mediating role of cultural capital:** Elderly rural individuals who engage in physical exercise gain access to valuable health knowledge and information, which helps mitigate multidimensional poverty. Those with higher cultural capital are typically better positioned to obtain exercise-related information through media, books, and the internet, and share it with their social circles, enhancing their understanding of health maintenance and disease prevention [64]. Additionally, individuals with higher cultural capital are more likely to engage in cultural activities that promote psychological well-being and quality of life [65]. These activities, such as mental health training and self-improvement courses, help break the poverty cycle. For example, elderly individuals who regularly participate in sports and cultural events often possess higher cultural literacy and social skills, which enables them to engage in volunteer work or health education, reducing the likelihood of multidimensional poverty for themselves and their families. Thus, elderly rural populations should view physical exercise as a means to engage with and shape cultural practices, such as traditional and folk sports, to enhance cultural capital and prevent both the onset and recurrence of poverty.

## 5.3. Heterogeneity analysis by gender, marital status, region, and age

The heterogeneity analysis reveals that physical exercise significantly impacts multidimensional poverty in elderly rural households, particularly for women, the unmarried, and those in eastern regions. **Gender**: Comparison of intergroup differences shows no significant variation in the core explanatory variable between male and female elderly groups. Previous research by Li et al. found that rural elderly men and women account for 1.88% and 3.76% of all impoverished populations, respectively [66]. Rural women face unequal access to education, employment, and social security, positioning them as marginalized and vulnerable within poverty-stricken groups [67]. Participation in physical exercise allows rural women to break free from traditional gender roles, enhancing their self-awareness and social status, while also promoting gender equality in rural areas. **Marital status**: Studies indicate that unmarried rural elderly individuals have a 5.581 percentage point higher incidence of multidimensional poverty compared to their married counterparts [5]. The lack of spousal support often leads to lower quality of life, mental poverty, and associated health issues. Physical exercise, through engagement in sports

clubs or other cultural activities, helps unmarried elderly individuals in rural areas improve subjective well-being and mental health. **Region**: The study finds that physical exercise has a more pronounced impact on reducing multidimensional poverty in elderly rural households in the eastern regions compared to the central and western regions, which contrasts with previous studies [68]. However, another study suggests that there is no inherent link between high economic development in eastern regions and lower multidimensional poverty indices, with some eastern provinces exhibiting higher poverty rates than many western regions [69]. This discrepancy may be explained by the larger agricultural production scale and lower social welfare in central and western rural areas, where elderly individuals are often engaged in farming or providing family labor. These factors, along with challenges in basic living needs, child-rearing, and healthcare access, hinder the effectiveness of physical exercise interventions in these regions. **Age:** The impact of physical exercise on multidimensional poverty is most pronounced among the oldest-old (aged 75 and above), with significant differences observed in intergroup comparisons. In this age group, accelerated physical decline and heightened risks of chronic illness and disability increase vulnerability to multiple poverty dimensions, particularly in health, income, and social participation. Health, therefore, becomes a foundational determinant of independence and resilience against multidimensional deprivation. Regular physical activity serves as a key intervention, offering multiple benefits: improved health status, reduced medical and caregiving costs, enhanced social engagement, and increased subjective well-being. These findings underscore the importance of developing and scaling age-appropriate, safe, and accessible community-based exercise programs tailored for the oldest-old, providing a critical pathway for alleviating multidimensional poverty in ageing rural populations.

## 6. Conclusion

This study explored the effects of physical activity participation on multidimensional poverty in old age among family farm households and its mechanisms, and the conclusions obtained are as follows: (1) The results of the OLS regression found that physical activity participation significantly reduced the probability of multidimensional poverty in old age among family farm households. Specifically, sustained participation in physical activity can promote health improvement, economic stability, improved living standards, and subjective feelings of goodness, which can provide an effective way for the rural elderly poverty group. (2) Mediation effect analyses using stepwise regression showed that social capital and cultural capital are important mediating mechanisms for physical activity participation in alleviating multidimensional poverty in old age among family farmers. (3) Heterogeneity results show that physical activity participation is particularly effective in reducing poverty among female household, unmarried, and elderly farmers in the eastern region, reflecting the differences in poverty reduction effects among groups with different characteristics.

Despite its contributions, this study has several limitations:(1) The CFPS variables may be subject to measurement error due to self-reported data, potentially leading to downward-biased estimates. Future research should incorporate multi-source data collection approaches to enhance the robustness of causal inference. (2) The CFPS dataset contains substantial missing values in the exercise duration variable, and the sample used is relatively limited. Future research should leverage larger and more diverse datasets for validation. (3) This study focuses on mediation and moderation mechanisms but does not explore the dynamic changes in the poverty-alleviation effects of exercise before and after China's targeted poverty alleviation campaign. Future work could adopt quasi-natural experimental designs and multi-period difference-in-differences (DID) models to address this gap. (4) The study targets elderly rural households, and the sample may not fully represent other regions or demographic groups. As rural revitalization and agricultural modernization advance, the generalizability of the findings warrants further verification. Future research should incorporate additional variables and utilize panel data to better identify causal pathways and enhance the robustness of conclusions.

The findings of this study offer important policy insights for addressing multidimensional poverty among elderly rural households in China. First, efforts should be made to enhance the construction and maintenance of rural sports infrastructure. As physical exercise plays a significant role in alleviating poverty, governments should increase investment in public fitness equipment, walking trails, and open spaces in rural areas. Special attention should be given to improving

the accessibility and usability of sports facilities in remote regions. In addition, an integrated system linking exercise and medical services should be established. This could include regular health screenings, exercise assessments, and real-time monitoring of elderly individuals' physical conditions to ensure effective and safe participation.

Second, a "healthy labour" model that integrates physical exercise with agricultural production should be developed. Many rural elderly individuals continue to engage in lifelong manual labour, both as a source of income and as a way to sustain self-worth. Introducing light exercise elements into agricultural activities—for example, promoting stretching, walking, or calisthenics during work—can help mitigate health risks associated with physical labour. This approach may enhance productivity while fostering long-term health–labour balance.

Third, targeted exercise-based poverty alleviation initiatives should be implemented for elderly populations at risk of falling back into poverty. To prevent poverty driven by illness or vulnerability, programs should focus on disadvantaged, frail, and isolated elderly individuals. These interventions can be combined with nutritional support, health education, and healthcare subsidies to strengthen the health component of poverty alleviation policy. Integration with existing elderly care institutions and home-based care services, alongside increased provision of community sports guidance and social support, can reduce both economic hardship and social exclusion caused by poor health.

Fourth, the "Healthy Village" initiative should be promoted to encourage universal participation in physical activity. Nationwide fitness campaigns hold significant potential for advancing rural revitalization. Through coordinated efforts by government, community organizations, and private enterprises, regular fitness events, health screenings, and educational programs can be organized. Community fitness challenges and incentive schemes may help mobilize elderly individuals and their families to adopt active lifestyles, enhancing awareness of senior health and fostering a culture of collective wellbeing in rural society.

## Author contributions

**Conceptualization:** Qifei Xia, Shu Xuan, Guo-you Qin.

**Data curation:** Qifei Xia, Shu Xuan, Guo-you Qin.

**Formal analysis:** Qifei Xia, Guo-you Qin.

**Methodology:** Guo-you Qin.

**Supervision:** Qifei Xia, Shu Xuan.

**Validation:** Qifei Xia.

**Writing – original draft:** Qifei Xia, Guo-you Qin.

**Writing – review & editing:** Qifei Xia, Shu Xuan, Guo-you Qin.

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
