## [Decision Letter · Decision Letter 0]

6 May 2025

Dear Dr. Qin,

Thank you for submitting your manuscript to PLOS ONE. After careful consideration, we feel that it has merit but does not fully meet PLOS ONE’s publication criteria as it currently stands. Therefore, we invite you to submit a revised version of the manuscript that addresses the points raised during the review process.

We look forward to receiving your revised manuscript.

Kind regards,

Enrico Ivaldi

Academic Editor

PLOS ONE

Journal Requirements:

2. Please remove all personal information, ensure that the data shared are in accordance with participant consent, and re-upload a fully anonymized data set.

Reviewers' comments:

Reviewer's Responses to Questions

**Comments to the Author**

1. Is the manuscript technically sound, and do the data support the conclusions?

Reviewer #1: Partly

Reviewer #2: Yes

2. Has the statistical analysis been performed appropriately and rigorously?

Reviewer #1: Yes

Reviewer #2: I Don't Know

3. Have the authors made all data underlying the findings in their manuscript fully available?

Reviewer #1: Yes

Reviewer #2: Yes

4. Is the manuscript presented in an intelligible fashion and written in standard English?

Reviewer #1: No

Reviewer #2: Yes

Reviewer #1: The study titled "Rural Active Aging as Development Strategy: Quantifying Exercise's Multidimensional Poverty Reduction Effects" uses the China Family Panel Studies (CFPS) adult and household databases and applies the A-F dual threshold method to measure and structurally decompose multidimensional poverty among elderly rural households.

The topic of poverty eradication—particularly multidimensional poverty encompassing access to health, education, and quality of life—is a top priority on the international development agenda. This effort is especially worthwhile among elderly rural households, a demographic that has rarely benefitted from empirical analysis.

However, there are several aspects that require significant improvement for the paper to be ready for publication. My main concern lies in the lack of compelling logic and an insufficiently established research gap. Specifically, the mechanism through which physical exercise influences multidimensional poverty is neither clearly articulated nor well emphasized, particularly regarding indirect effects. This relegates the study to a more academic exercise rather than a practically meaningful inquiry. Furthermore, the nature of physical exercise included in the analysis is unclear—why is it captured as a broad dummy variable? Given the focus on elderly rural populations, how much exercise is realistically impactful in alleviating poverty?

Below are my specific comments and suggestions:

Abstract

The abstract is well-written but lacks focus in logically presenting the research findings and recommendations. For example:

" Physical exercise has a significant poverty-reducing effect, with a multifaceted impact….."

This statement is too vague. How does physical exercise reduce poverty, and in what ways is the impact multifaceted? This is not sufficiently addressed, leaving the reader uncertain about what aspects of poverty are being discussed. The authors must explicitly state the pathway through which physical exercise contributes to multidimensional poverty reduction.

Introduction

The introduction presents a strong case but does not clearly articulate the knowledge gap. What is the central problem? Are elderly individuals not engaging in exercise, or is multidimensional poverty particularly pronounced among them? Could it be that multidimensional poverty itself affects exercise frequency? Furthermore, why are internet access and social capital not considered? All elements of the theoretical framework should be integrated into this section to demonstrate how they logically connect to the research problem.

Literature Review

What is the theoretical foundation of the study? Upon what framework are the hypotheses built? The authors appear to rely primarily on empirical literature rather than an established theory that supports the idea that exercise can reduce multidimensional poverty. Additionally, the mediating and moderating variables should be theoretically justified.

Data

A key issue is how physical exercise is operationalized. Considering the study focuses on elderly rural populations, using a broad dummy variable may not be appropriate, especially since the type of exercise is unspecified. It is crucial to clarify what kinds of exercise are included and how much is required to have a meaningful effect. Additionally, what are the criteria for exercise in terms of age groups? A more detailed description of the sample is warranted.

Regarding the Multidimensional Poverty Index (MPI), beyond the aggregate measure, the authors should disaggregate poverty into its key components—health, education, and quality of life—to ensure the robustness of the results.

The empirical strategy should be made more reader-friendly. Complex methodologies need to be explained clearly for broader accessibility. Further methodological concerns include the treatment of endogeneity and reverse causality—how were these addressed? Additionally, it is recommended to bootstrap the Sobel test for more accurate standard errors, which is a crucial step in ensuring the validity of the findings.

Results

The presentation of results is unclear and confusing.

Each model should be explained separately, including a detailed explanation of the columns in each results table.

The findings should be interpreted within the context of existing literature to highlight contributions and deviations.

Most critically, the causal pathway between physical exercise and multidimensional poverty must be better established. Otherwise, the findings risk appearing too simplistic.

Conclusion

The conclusion is notably weak. It lacks practical policy recommendations and fails to acknowledge the study's limitations or suggest directions for future research.

A strong conclusion would:

• Emphasize the practical implications of promoting physical exercise among elderly rural populations,

• Discuss the feasibility of interventions,

• And acknowledge constraints such as cultural factors, health limitations, or infrastructure gaps.

Reviewer #2: The paper 'Rural Active Aging as Development Strategy: Quantifying Exercise's Multidimensional Poverty Reduction Effects', addresses the issue of rural poverty, which is becoming increasingly multidimensional and is an important yet often overlooked aspect of poverty. It scientifically evaluates the relationship between physical activity and multidimensional poverty among rural elderly households in China in order to consolidate the results of poverty alleviation and effectively link them to rural revitalisation. Using data from the China Family Panel Studies (CFPS) adult databases from 2018 and 2022, the paper calculates the Multidimensional Poverty Index (MPI), considering four dimensions — health status, economic conditions, living standards and subjective well-being — and 12 specific indicators. This paper makes an important contribution to research by addressing a timely topic and providing decision-makers with interesting insights. It is well written, the English is appropriate, the structure is clear and the discussion is consistent with the literature. Nevertheless, I would like to suggest some minor revisions that could improve the paper:

- The literature in the Lit. Rev. section should be updated further. Some citations are quite dated.

- 'Limitations and future agenda' should be expanded upon further and moved to the 'Conclusions' section.

- In order to strengthen generalisability, I would expand the reflection on why one would expect the effects to vary in other populations (e.g. young people or urban populations) or geographical contexts (e.g. countries with different welfare systems, sports infrastructures, or cultural norms regarding exercise).

For this reason, I have assigned minor revisions to this paper.

**Do you want your identity to be public for this peer review?** For information about this choice, including consent withdrawal, please see our Privacy Policy

Reviewer #1: No

Reviewer #2: No

---

## [Author Response · Author response to Decision Letter 1]

26 May 2025

Rural Active Aging as Development Strategy: Quantifying Exercise’s Multidimensional Poverty Reduction Effects

Qi-fei Xia1, Shu Xuan2, Guo-you Qin3*

1 School of Physical Education, Ankang College, Ankang, China

2 School of Physical Education, Hunan University, Changsha, Hunan, China

3College of Physical Education, Hanjiang Normal University, Shiyan, China

* Correspondence: Prof. Guo-you Qin,

Email: qinguoyou22@163.com

Response Letter to Reviewer’s Comments

Comments from the Editor:

Thank you for submitting your manuscript to PLOS ONE. After careful consideration, we feel that it has merit but does not fully meet PLOS ONE’s publication criteria as it currently stands. Therefore, we invite you to submit a revised version of the manuscript that addresses the points raised during the review process. The reply letter has been uploaded as an attachment.

Reponse: We thank all the editors and reviewers for their valuable comments and suggestions.-We have carefully revised the manuscript to enhance its clarity and facilitate the understanding of the readers. Our point-to-point responses are presented in the following We-hope that the revision would satisfactorily address the comments and concerns of the editors and reviewers.

Response Letter

Dear Editor:

Thank you for your letter and comments concerning our manuscript entitled “Rural Active Aging as Development Strategy: Quantifying Exercise’s Multidimensional Poverty Reduction Effects” (Manuscript ID: PONE-D-25-1951), Those comments are all valuable and very helpful for revising and improving our paper. According to the Editor/Reviewer’s comments. we have made extensive modifications to our manuscript. In this revised version. Our response is given in normal front and changes/additions to the manuscript are given in the red text. the detailed corrections are listed below. The reply letter has been uploaded as an attachment.

I hope that the revised manuscript for publication in PLOS One. Thank you again for your help with the manuscript. I am more than happy to provide any further information you may need.

---

## [Decision Letter · Decision Letter 1]

12 Jun 2025

Dear Dr. Qin,

Thank you for submitting your manuscript to PLOS ONE. After careful consideration, we feel that it has merit but does not fully meet PLOS ONE’s publication criteria as it currently stands. Therefore, we invite you to submit a revised version of the manuscript that addresses the points raised during the review process.

We look forward to receiving your revised manuscript.

Kind regards,

Enrico Ivaldi

Academic Editor

PLOS ONE

Reviewers' comments:

Reviewer's Responses to Questions

**Comments to the Author**

Reviewer #1: (No Response)

Reviewer #2: All comments have been addressed

2. Is the manuscript technically sound, and do the data support the conclusions?

Reviewer #1: Partly

Reviewer #2: Yes

3. Has the statistical analysis been performed appropriately and rigorously?

Reviewer #1: No

Reviewer #2: Yes

4. Have the authors made all data underlying the findings in their manuscript fully available?

Reviewer #1: Yes

Reviewer #2: Yes

5. Is the manuscript presented in an intelligible fashion and written in standard English?

Reviewer #1: Yes

Reviewer #2: Yes

Reviewer #1: The paper has improved in several areas compared to the earlier version; however, there are still significant issues that need to be addressed before it can be considered for publication.

Abstract

The abstract requires a complete rewrite. The current structure, which follows subheadings like Introduction, Methods, Results, Conclusion, and Recommendations, is not appropriate for this journal. Moreover, Propensity Score Matching (PSM) is mentioned under results but not reflected in the methos. All methods used should be clearly outlined along with their purpose and relevance to the study.

Additionally, there are inconsistencies in the figures presented for the indirect effects between the document and the submission system. Most importantly, the results section lacks clarity. Only results directly linked to the study’s objectives should be presented; currently, the inclusion of mediating factors, moderating variables, and heterogeneous effects lacks coherence and alignment. As it stands, the presentation in the abstract is unclear and disjointed without clearly showing the mechanism.

Introduction

This version of the introduction is more refined than the previous submission. However, there are still significant gaps:

Economic Rationale of Physical Exercise: The paper does not adequately explain how physical exercise contributes to income premiums. There is a need to clarify whether physical exercise incurs financial costs—potentially reducing disposable income for the elderly—or whether it is a free public service that can improve health and reduce medical expenditures. This nuance is essential in understanding the role of exercise in addressing multidimensional poverty, which extends beyond health to encompass education and overall quality of life. The authors are encouraged to reference relevant literature to build a stronger argument for this connection. Moreover, Are the exercises studied communal or individual in nature?

Policy Context – National Fitness and Healthy China Strategies: A brief but clear elaboration on the National Fitness and Healthy China strategies would enhance the introduction by grounding the study within the national policy framework. This background is critical for understanding why physical exercise is a focus in the first place and how it ties into broader goals of poverty reduction and well-being.

Mediating and Moderating Variables – Contextual Clarity: The roles of social capital (the value of an individual's position within organizational structures) and cultural capital (the precondition for engaging in aesthetic and recreational activities, often tied to educational background) are introduced, but their relevance is not clearly explained. Key questions remain unanswered: What is the mechanism through which these forms of capital mediate the relationship between physical exercise and multidimensional poverty? Similarly, the moderating roles of internet access and public service provision must be grounded in a contextual framework. It is not enough to say these factors were previously unconsidered; authors must justify their inclusion based on theoretical or empirical reasoning, showing how these variables interact with the core relationship under study. Context and conceptual connection are essential for establishing academic and practical relevance. In short why are they important in this study? What is the background?

Literature Review

The focus on multidimensional poverty is appreciated, but the literature review leans heavily toward public physical exercise, a focus not clearly emphasized earlier in the manuscript. Are the authors advocating for public/community physical exercise, or is individual exercise also of interest? For instance, how does individual exercise promote social capital in the Chinese rural context?

Additionally, the use of internet services is vague. In Hypothesis H3a, it is unclear whether the internet is used for accessing exercise programs, purchasing equipment, or social networking. This lack of clarity weakens the theoretical foundation.

Data and Sampling

The decision to define elderly rural residents as individuals aged 45 and above needs stronger justification. Why not 50 or 55? Those aged 45 may have very different physical capacities and socioeconomic contexts compared to those 70 and above. The empirical strategy must consider these age-related differences, especially given that mediating and moderating effects may differ substantially by age cohort.

Variables and Methodology

• Dependent Variable: There is inconsistency in the stated weights applied across the four dimensions of multidimensional poverty. The text claims equal weights, but Table 1 presents unequal weights (1/16 and 1/8). This discrepancy must be resolved.

• Community Social Capital: Authors should clarify how the composite indicator was transformed into a binary variable and why both individual and community levels of social capital are used.

• Equation 12: The interaction term is discussed in the text but not shown in the actual equation. Furthermore, the manuscript lacks clear specification of the models used to estimate the relationships. A dedicated section explaining model choice and their suitability is necessary.

• Model Specification: The manuscript lacks a clear presentation of the models used to estimate the relationships. This is a major omission. Authors must dedicate a section to describe the econometric models applied and explain why they are appropriate. Robustness checks and treatment of endogeneity should also be thoroughly discussed.

Results

This section has major conceptual and methodological weaknesses that compromise the credibility of the findings.

Lack of Model Specification: The authors begin the results section without stating what model(s) were used for the initial estimation. It is unclear whether an OLS, probit, logit, or another specification was applied in the first stage of the analysis. The term “baseline model” is mentioned but not adequately defined, nor is its function in the analysis (i.e., whether it serves as the main specification or as a robustness check). This must be corrected. Without clearly presenting the baseline econometric model, its control variables, and justification, the rest of the analysis becomes difficult to interpret or trust.

Endogeneity and Self-selection Bias Not Addressed: Participation in physical exercise is likely endogenous, as individuals who choose to participate may differ systematically from non-participants in both observable and unobservable characteristics . The paper does not explain how it addresses self-selection bias. While Propensity Score Matching (PSM) is briefly mentioned elsewhere, it is not integrated coherently into the results narrative. If PSM or any other method (e.g., Heckman selection model, IV approach) was used, it must be formally introduced, justified, and connected to the tables showing the main results. Otherwise, the estimates are at high risk of bias.

Inappropriate and Incomplete Mediation Analysis (Sobel Test): The use of the Sobel test is methodologically flawed: First, the Sobel test assumes a single mediator in the causal pathway. The authors are using two mediators—social capital and cultural capital—which violates this assumption. The Sobel test does not extend to parallel or sequential multiple mediation models, and using it in this context is statistically inappropriate. Therefore, the reported mediation results are highly questionable. Second, even within the (misapplied) Sobel framework, the authors fail to provide the total, direct, and indirect effects for each mediator. These values are critical for interpreting the mediation mechanism. It is unclear how much of the effect of physical exercise on multidimensional poverty is transmitted via each mediator, nor whether the indirect effects are statistically significant.There is also no discussion of the assumptions underlying this test or any robustness checks for the mediation pathways.

4. Ambiguity Around KHB Decomposition: The authors mention using KHB decomposition, but the presentation lacks transparency and technical rigor: It is unclear how the estimates were derived, what specific models were decomposed, and whether the KHB method was implemented properly given the type of outcome variable used. There is no citation or methodological explanation of the KHB procedure. The authors must specify the model used for KHB decomposition, the variables included, and the interpretation of the decomposed components (confounding, mediation, etc.). Most critically, there is no clear distinction between what the Sobel test results contribute and what KHB decomposition adds. If both methods were used, the rationale for this dual approach must be stated and justified. If KHB replaced the Sobel test, the latter should not be reported at all. The authors are strongly encouraged to revise this section in light of the methodological shortcomings. For proper guidance, they should consult the following methodological literature on mediation analysis and KHB decomposition:

Without addressing these concerns, the validity of the mediation and moderation results remains severely undermined. The authors must ensure the results presented are methodologically sound, well-documented, and interpretable by scholars.

Conclusion

While this section is more refined than previous versions, the limitations are still not sufficiently addressed. The issue of self-reported responses should be acknowledged as a potential source of limitation.

Reviewer #2: The authors have addressed all comments indicated. The paper is ready for publication in this esteemed Journal by my side.

**Do you want your identity to be public for this peer review?** For information about this choice, including consent withdrawal, please see our Privacy Policy

Reviewer #1: No

Reviewer #2: No

---

## [Author Response · Author response to Decision Letter 2]

16 Jun 2025

Dear Editor/Reviewer:

Thank you for your letter and comments concerning our manuscript entitled “Rural Active Aging as Development Strategy: Quantifying Exercise’s Multidimensional Poverty Reduction Effects” (Manuscript ID: PONE-D-25-1951), Those comments are all valuable and very helpful for revising and improving our paper. According to the Editor/Reviewer’s comments. we have made extensive modifications to our manuscript. In this revised version. Our response is given in normal front and changes/additions to the manuscript are given in the red text. the detailed corrections are listed below. The letter of reply has been placed in the annex.

I hope that the revised manuscript for publication in PLOS One. Thank you again for your help with the manuscript. I am more than happy to provide any further information you may need.

---

## [Decision Letter · Decision Letter 2]

20 Jun 2025

Dear Dr. Qin,

Thank you for submitting your manuscript to PLOS ONE. After careful consideration, we feel that it has merit but does not fully meet PLOS ONE’s publication criteria as it currently stands. Therefore, we invite you to submit a revised version of the manuscript that addresses the points raised during the review process.

We look forward to receiving your revised manuscript.

Kind regards,

Enrico Ivaldi

Academic Editor

PLOS ONE

Additional Editor Comments:

the rew complains that most of its directions have not been met. I ask the authors to pay close attention to what is requested and to respond point by point

Reviewers' comments:

Reviewer's Responses to Questions

**Comments to the Author**

Reviewer #1: (No Response)

Reviewer #2: All comments have been addressed

2. Is the manuscript technically sound, and do the data support the conclusions?

Reviewer #1: Partly

Reviewer #2: Yes

3. Has the statistical analysis been performed appropriately and rigorously?

Reviewer #1: No

Reviewer #2: Yes

4. Have the authors made all data underlying the findings in their manuscript fully available?

Reviewer #1: Yes

Reviewer #2: Yes

5. Is the manuscript presented in an intelligible fashion and written in standard English?

Reviewer #1: No

Reviewer #2: Yes

Reviewer #1: Thank you for your revised submission. While the subject of your study remains relevant, I am disappointed that many of my earlier comments were not sufficiently addressed. Please find my detailed comments attached. I strongly encourage you to review both the original feedback and the additional points provided in light of your responses. Greater attention to detail and a more careful engagement with the suggestions will be important for preparing a publication-ready manuscript.

Reviewer #2: The authors have properly addressed all the indications provided by the reviewers. This paper is ready for publication.

**Do you want your identity to be public for this peer review?** For information about this choice, including consent withdrawal, please see our Privacy Policy

Reviewer #1: No

Reviewer #2: No

---

## [Author Response · Author response to Decision Letter 3]

21 Jun 2025

Dear Editor

Thank you for your letter and comments on our manuscript entitled ‘Positive Rural Aging as a Development Strategy’ (Manuscript No. PONE-D-25-1951), which have been helpful in revising and improving the paper. We have always found the reviewers' suggestions helpful in improving the quality of the manuscript, and we have uploaded the revised responses as an attachment in the hope that our revisions will allay the reviewers' concerns. Meanwhile, we are very grateful to the reviewers for giving us the opportunity to revise the manuscript, which makes us feel the warmth of your rigorous working attitude, and we will continue to provide more valuable research results. We have put the manuscript revision responses and revised manuscript in the attachment. We wish you a happy life and good luck in your work.

---

## [Decision Letter · Decision Letter 3]

11 Jul 2025

Dear Dr. Qin,

Thank you for submitting your manuscript to PLOS ONE. After careful consideration, we feel that it has merit but does not fully meet PLOS ONE’s publication criteria as it currently stands. Therefore, we invite you to submit a revised version of the manuscript that addresses the points raised during the review process.

We look forward to receiving your revised manuscript.

Kind regards,

Enrico Ivaldi

Academic Editor

PLOS ONE

Journal Requirements:

Additional Editor Comments:

Dear Authors,

We kindly ask you to provide a detailed and thorough response to the comments raised by the reviewer in their latest evaluation. It is essential to carefully address all the points highlighted, making the necessary revisions and offering comprehensive clarifications.

This approach will significantly improve the quality of your work and increase the likelihood of its unconditional acceptance.

Please feel free to contact us if you require any further clarification. Thank you for your cooperation.

Reviewers' comments:

Reviewer's Responses to Questions

**Comments to the Author**

Reviewer #1: (No Response)

2. Is the manuscript technically sound, and do the data support the conclusions?

Reviewer #1: (No Response)

3. Has the statistical analysis been performed appropriately and rigorously?

Reviewer #1: Yes

4. Have the authors made all data underlying the findings in their manuscript fully available?

Reviewer #1: Yes

5. Is the manuscript presented in an intelligible fashion and written in standard English?

Reviewer #1: No

Reviewer #1: Dear Authors,

Thank you for submitting the revised version of the manuscript. However, I find that the revisions appear rather rushed, and there remains a lack of careful attention to my previous comments. It is always better to fully understand each comment and address it thoroughly, rather than provide partial revisions or vague rebuttals. I strongly encourage you to carefully consider each point and either fully address the comment or provide a clear and explicit rebuttal where necessary.

One of the main concerns with the current revision is that you continue adding elements that were not requested, in an attempt to justify your initial approach, instead of making the required changes. In the interest of time and clarity, I will be very direct in my comments below. In your next response, I expect you to explicitly state whether each point has been addressed or not, without lengthy or unfocused explanations. Simply state how you addressed the point, or why you chose not to, in a succinct and focused manner.

Title Recommendation:

The revised title should not include the phrase “An Empirical Analysis Based on CFPS Data”, especially since most readers may not be familiar with CFPS. Below are two conditional title suggestions, depending on your chosen methodological approach:

If you decide to use an IV method like PSM, with a well-identified instrumental variable as the main model to estimate the effect of physical exercise on multidimensional poverty (and use the baseline model only for robustness checks):

Suggested Title:

Rural Active Aging as a Development Strategy: Impact of Physical Activity Participation on Multidimensional Poverty in China

If you choose to retain the baseline regression as your main model and remove the robustness checks conducted using the IV approach (which, in my view, is more methodologically sound):

Suggested Title:

Rural Active Aging as a Development Strategy: Nexus of Physical Activity Participation and Multidimensional Poverty in China

Note: Please carefully consider whether “physical activity” or “physical exercise” is the most appropriate term for your study. I believe physical exercise may be more accurate.

Conceptual Framing and Hypotheses:

I recommend that the conceptual framing focuses on physical exercise and the mediating role you intend to test. Given the limited and unclear data coverage of public service and internet use variables, it would be better to avoid emphasizing these as moderators in your hypotheses.

Specifically, Hypothesis 31a remains unclear. The term “internet use” is too vague, lacking clarity on what aspects of internet use are being tested. Hypotheses must be specific and testable—not included for the sake of it. I strongly suggest removing this hypothesis. However, you may still analyze this moderating effect in the results section, provided it is carefully presented in the methods.

Methods Section Structuring:

Here’s how you could improve the methods section under both scenarios:

If following Option 2 (Baseline approach):

2.2 Analytical Strategy

2.2.1 OLS:

Explain how you use OLS to analyze the association between physical exercise and multidimensional poverty. Briefly describe why OLS is appropriate. Additionally, explain how interaction terms (e.g., for internet use or public services) were incorporated to test for moderation, if you choose to retain this analysis.

2.2.2 Stepwise regression or KHB decomposition:

You should select only one of these techniques. Clearly justify your choice in a brief and intelligent manner. There is no need for a dedicated subsection to explain this—just integrate the rationale within the methods.

This structure will make your methods section more focused and clearer.

If following Option 1 (IV/PSM approach):

2.2.1 PSM or IV Method:

Select a valid instrumental variable from the dataset—one that affects physical exercise participation but not multidimensional poverty directly. This will help isolate the causal impact of physical exercise.

2.2.2 Moderating Effects:

Use a similar OLS-based interaction approach as suggested above for testing moderation effects.

2.2.3 Robustness Checks:

Include OLS models and moderation analyses here as part of your robustness checks. This structure would greatly strengthen the methodological rigor and overall focus of the paper.

Results and Conclusion:

The results section should logically follow the selected methodological approach, and the conclusion should directly reflect the key findings and chosen analytical strategy.

Thank you for your efforts so far. I look forward to a thoroughly revised and carefully considered next submission.

**Do you want your identity to be public for this peer review?** For information about this choice, including consent withdrawal, please see our Privacy Policy

Reviewer #1: No

---

## [Author Response · Author response to Decision Letter 4]

13 Jul 2025

Dear Reviewer:

Thank you for your letter and comments concerning our manuscript entitled “Rural Active Aging as Development Strategy: Quantifying Exercise’s Multidimensional Poverty Reduction Effects” (Manuscript ID: PONE-D-25-1951), Those comments are all valuable and very helpful for revising and improving our paper. We are very sorry that we have not been able to address your concerns well in each revision, and in this revision process we have carefully revised all of the sections from the Abstract, Methodological Strategy, and Results and Discussion sections. We would like to explain that we think about revision options immediately after receiving your suggestions in each case, and this manuscript is crucial for us because we need to conduct title reviews or other similar studies this year, which makes us respond more quickly or hurriedly. But don't worry, we have reviewed a lot of literature during each revision process, and we are very grateful for your professional advice, which has helped us to improve the quality of the manuscript, after we have given it much thought and hope to reply to you in the fastest time possible. In order to better complete this revision, we have provided a revised version (Word) version of the manuscript, and we have also added revised annotations inside the manuscript, so that you can better access, and we have also provided a letter of reply to your revised comments, according to your requirements, briefly explaining the contents of the revision, and hope to solve your concerns. Thank you again for your review. The letter of reply has been placed in the annex.

I hope that the revised manuscript for publication in PLOS One. Thank you again for your help with the manuscript. I am more than happy to provide any further information you may need.

---

## [Decision Letter · Decision Letter 4]

13 Aug 2025

Rural Active Aging as a Development Strategy: Nexus of Physical Exercise Participation and Multidimensional Poverty in China

PONE-D-25-12951R4

Dear Dr. Qin,

We’re pleased to inform you that your manuscript has been judged scientifically suitable for publication and will be formally accepted for publication once it meets all outstanding technical requirements.

Kind regards,

Enrico Ivaldi

Academic Editor

PLOS ONE

Additional Editor Comments (optional):

Reviewers' comments:

Reviewer's Responses to Questions

**Comments to the Author**

Reviewer #1: All comments have been addressed

2. Is the manuscript technically sound, and do the data support the conclusions?

Reviewer #1: Yes

3. Has the statistical analysis been performed appropriately and rigorously?

Reviewer #1: Yes

4. Have the authors made all data underlying the findings in their manuscript fully available?

Reviewer #1: Yes

5. Is the manuscript presented in an intelligible fashion and written in standard English?

Reviewer #1: Yes

Reviewer #1: Dear Authors,

Thank you for taking the time to thoroughly address the review comments provided. I have carefully gone through the revised manuscript, and I must commend you for the thoughtful revisions and enhancements made in response to the feedback.

The clarity of the theoretical framework has improved, and your deeper engagement with the mediating mechanisms has added valuable nuance to the discussion. Additionally, the integration of methodological limitations into the main discussion section contributes to a more balanced and transparent presentation of your findings.

Overall, the quality and coherence of the paper have significantly improved, and the manuscript is now much stronger as a result of your diligent efforts.

**Do you want your identity to be public for this peer review?** For information about this choice, including consent withdrawal, please see our Privacy Policy

Reviewer #1: No

---

## [Editor Report · Acceptance letter]

PONE-D-25-12951R4

PLOS ONE

Dear Dr. Qin,

I'm pleased to inform you that your manuscript has been deemed suitable for publication in PLOS ONE. Congratulations! Your manuscript is now being handed over to our production team.

Kind regards,

on behalf of

Prof. Enrico Ivaldi

Academic Editor

PLOS ONE